# Information Leakage Rate of Optical Code Division Multiple Access Network Using Wiretap Code

**DOI:** 10.3390/e25101384

**Published:** 2023-09-26

**Authors:** Rongwo Xu, Leiming Sun, Jianhua Ji, Ke Wang, Yufeng Song

**Affiliations:** College of Electronics and Information Engineering, Shenzhen University, Shenzhen 518060, China; 2100432075@email.szu.edu.cn (R.X.); 2350434003@email.szu.edu.cn (L.S.); wong@szu.edu.cn (K.W.); yfsong@szu.edu.cn (Y.S.)

**Keywords:** optical CDMA, physical layer security, secrecy capacity, information leakage rate

## Abstract

Secrecy capacity is usually employed as the performance metric of the physical layer security in fiber-optic wiretap channels. However, secrecy capacity can only qualitatively evaluate the physical layer security, and it cannot quantitatively evaluate the physical layer security of an imperfect security system. Furthermore, secrecy capacity cannot quantitatively evaluate the amount of information leakage to the eavesdropper. Based on the channel model of an optical CDMA network using wiretap code, the information leakage rate is analyzed to evaluate the physical layer security. The numerical results show that the information leakage rate can quantitatively evaluate the physical layer security of an optical CDMA wiretap channel, and it is related to transmission distance, eavesdropping position, confidential information rate and optical code.

## 1. Introduction

Fiber-optic communication systems are vulnerable to various types of physical layer attacks [1]. For example, an eavesdropper (Eve) can extract a portion of the transmitted signal by bending the fiber. In this way, Eve can recover the original signal and cannot be detected easily by legitimate users [2]. Quantum key distribution (QKD) can theoretically provide absolute security, but it cannot support high-speed data streams, such as a key rate of only 0.014 bps under a 833 km optical fiber transmission distance [3].

The physical layer security of fiber-optic networks is increasingly important because it guarantees the confidentiality of information without compromising the computing power of Eve, and it eliminates the key distribution and management required by traditional encryption techniques [4,5,6,7]. In 1975, Wyner proposed the wiretap channel model and secrecy capacity [8]. Secrecy capacity was defined as the maximum achievable system transmission rate at which Eve could not gain any useful information about a message. Later, it was extended to broadcast channels with confidential messages and Gaussian wiretap channels [9,10]. Kyle Guan et al. analyzed the security of space-division multiplexed fiber-optic communication systems and used distortion as a quantitative metric for secrecy. They investigated how the rate of reliable communication between the legitimate transmitter–receiver pair could be chosen to maximize reconstruction errors of Eve [11]. Then, they further analyzed the information theoretic security of multiple-input–multiple-output space-division multiplexed fiber-optic communication systems in the presence of multiple Eves [12]. Physical layer security is an important performance parameter of optical code division multiple access (OCDMA), which can improve the security of optical fiber transmission systems [13]. Yeteng Tan et al. proposed a novel secure communication scheme based on OCDMA technology, and secrecy capacity was employed to evaluate the physical layer security level [14]. The secrecy capacity of a quantum secure direct communication system was studied [15]. Andrew Lonnstrom et al. proposed a method for optimizing the information theoretic secure goodput of a multiple-input–multiple-output degraded wiretap channel using inverse precoding [16]. In order to quantify the security of specific coding schemes, the rate distortion of multimode optical fiber communication systems was studied [17].

A wiretap code can be designed by choosing two code rates, namely, the codeword rate *R_b_* and the rate of transmitted confidential information *R_s_* [18]. The redundancy rate *R_e_* = *R_b_* − *R_s_* is used to confuse Eve. In order to ensure the reliability and security, we must ensure *R_b_* ≤ *C_B_* and *R_e_* ≥ *C_E_*, where *C_B_* and *C_E_* are the main channel capacity and the wiretap channel capacity, respectively. When the main channel is better than the wiretap channel, secrecy capacity is defined as *C_S_* = *C_B_* − *C_E_*, which is the difference between the main channel capacity and the wiretap channel capacity. Secure communication can be achieved as long as *R_s_ ≤ C_S_*.

However, for a long-distance fiber-optic communication system, when Eve is close to the transmitter, Eve can obtain a higher signal-to-noise ratio (SNR) than the legitimate users. Although we can reduce the channel capacity of Eve by sending artificial noise [19], the system still cannot guarantee the physical layer security. On the other hand, because Eve’s location is unknown, there are several different cases of security in the whole communication link. (1) *C_S_ ≥ R_s_*: in this case, the communication system can achieve perfect security. (2) 0 *< C_S_ < R_s_*: in this case, some confidential information is leaked to Eve. (3) *C_S_ =* 0: in this case, Eve can obtain all the confidential information. Therefore, secrecy capacity can only qualitatively evaluate the physical layer security under perfect security conditions; it cannot quantitatively evaluate the physical layer security under imperfect security conditions. Moreover, secrecy capacity cannot quantitatively evaluate the information leakage.

In this paper, we investigate the information leakage rate of an OCDMA network using wiretap code. The rest of this paper is organized as follows: In Section 2, we propose the channel model of the OCDMA network using wiretap code and theoretically analyze the information leakage rate of the fixed-rate wiretap code. The numerical results and discussions will be given in Section 3. The conclusion will be given in Section 4.

## 2. System Model and Theoretical Analysis

Figure 1 depicts the channel model based on OCDMA using wiretap code. At the transmitter, Alice outputs confidential information *M*, which is encoded by a wiretap channel encoder and an OCDMA encoder. Then, an *n*-vector *X^n^* is transmitted through the fiber channel. The length of the fiber link is *L*, and the legitimate user Bob receives *Y^n^*. After using a matched OCDMA decoder and optically amplified receiver (OAR), Bob can recover confidential information via the wiretap channel decoder. On the other hand, Eve intends to obtain useful information at extraction location *l* with an extraction ratio *x*. According to *Kerckhoffs*’s principle [20], Eve knows the data rate, coding type and code length, but does not know the specific optical code used by the legitimate user. Hence, Eve receives *Z^n^* and can only use an unmatched OCDMA decoder. The wiretap system adopts random coding. The confidential information rate is $R_{s}=H(M)/n$, where *H*(*M*) is the entropy of *M*. The codeword rate is $R_{b}=H(X^{n})/n$, where *H*(*X^n^*) is the entropy of *X^n^*. The wiretap code is constructed by generating $2^{nR_{b}}$ codewords. For each message $u=\left\{1,2,3,\cdots ,2^{nR_{s}}\right\}$, we randomly select one codeword from $v=\left\{1,2,3,\cdots ,2^{n(R_{b}-R_{s})}\right\}$. The OCDMA encoder uses optical orthogonal code (*F*, *W*, 1, 1), where *F* is the code length, *W* is the code weight and the autocorrelation and cross-correlation limits are 1.

At the receiver, OAR includes an erbium-doped fiber amplifier (EDFA), an optical filter, a photodiode, a low-pass electrical filter (LPF) and a decision circuit, as shown in Figure 2. In this system, the noise mainly includes shot noise, signal–spontaneous beat noise, spontaneous–spontaneous beat noise, thermal noise and dark current noise.

At the transmitter, *P* is the optical power of chip “1” after the OCDMA encoder. The optical fiber attenuation coefficient is *ɑ*, and the chip power *P_l_* at the position *l* is [21]
(1)$$P_{l}=\frac{P}{10^{\alpha l/10}}$$

Because Eve adopts an unmatched decoder, the cross-correlation value of optical orthogonal code is 1. Therefore, the optical power of Eve’s receiver is
(2)$$P_{Eve}=xP_{l}$$

The legitimate user adopts a matched decoder, and the autocorrelation peak value of optical orthogonal code is *W*. Hence, the optical power of Bob’s receiver is
(3)$$P_{S}=(1-x)\frac{WP}{10^{\alpha L/10}}$$

When the user data are “1”, the mean current value and the total noise are expressed as [22]
(4)$$I_{m1}=R_{c}(GP_{S}+P_{ASE})$$
(5)$$\begin{array}{ll} \sigma_{m1}^{2} & =\sigma_{sh1}^{2}+\sigma_{s\text{-}sp1}^{2}+\sigma_{\mathrm{sp}\text{-}\mathrm{sp}}^{2}+\sigma_{\mathrm{th}}^{2}+\sigma_{d}^{2}\\ & =2eB_{e}R_{c}(GP_{S}+P_{ASE})+2\frac{B_{e}}{B_{o}}GR_{c}^{2}P_{S}P_{ASE}+\frac{B_{e}}{B_{o}^{2}}{\left(R_{c}P_{ASE}\right)}^{2}(2B_{o}-B_{e})\\ & +(4k_{B}T/R)B_{e}+2eI_{d}B_{e} \end{array}$$
where *R_c_* is the receiver responsivity, *B_o_* is the optical bandwidth, *G* is the amplifier gain, *e* is electron charge and *N_sp_* is the spontaneous radiation factor. Amplified spontaneous radiation noise power is $P_{ASE}=2hvN_{sp}(G-1)B_{o}$ and *R* is the receiver load resistance. Photodetector bandwidth $B_{e}=(3/4)FR_{b}$, *T* is the temperature, *k_B_* is Boltzmann constant and *I_d_* is dark current.

When the user data are “0”, the mean current value and the total noise are expressed as [22]
(6)$$I_{m0}=R_{c}P_{ASE}$$
(7)$$\begin{array}{ll} \sigma_{m0}^{2} & =\sigma_{sh0}^{2}+\sigma_{\mathrm{sp}\text{-}\mathrm{sp}}^{2}+\sigma_{\mathrm{th}}^{2}+\sigma_{d}^{2}\\ & =2eB_{e}R_{c}P_{ASE}+2\frac{B_{e}}{B_{o}}GR_{c}^{2}P_{S}P_{ASE}+\frac{B_{e}}{B_{o}^{2}}{\left(R_{c}P_{ASE}\right)}^{2}(2B_{o}-B_{e})\\ & +(4k_{B}T/R)B_{e}+2eI_{d}B_{e} \end{array}$$

The bit error rate (BER) of legitimate users can be calculated by
(8)$$P_{e}=\frac{1}{2}erfc(\frac{Q}{\sqrt{2}})\approx \frac{\exp(-Q^{2}/2)}{Q\sqrt{2\pi }}$$
where $Q=(I_{m1}-I_{m0})/(\sigma_{m1}+\sigma_{m0})$, *erfc*() is the complementary error function. Similarly, Eve’s BER can be calculated.

It is assumed that the probability of user data being “0” and “1” is equal, and the channel is simplified to a binary symmetric channel model with an error transmission probability *P_e_*, as shown in Figure 3.

The main channel capacity is
(9)$$C_{B}=\max\left\{I(X^{n};Y^{n})\right\}=1-h(P_{e})$$
where $h(P_{e})=-P_{e}\log(P_{e})-(1-P_{e})\log(1-P_{e})$. After calculating the BER of Eve, the channel capacity of Eve can also be obtained.

Partial secrecy is usually quantified by the equivocation. In this paper, we use fractional equivocation, which is defined as [10]
(10)$$\Delta =\frac{H(M|Z^{n})}{H(M)}$$

*H*(*M*|*Z^n^*) is denoted as the entropy of residual uncertainty at Eve. Therefore, in a partial secrecy scenario, the maximum achievable fractional equivocation can be obtained from the following equation [23]:(11)$$\Delta =\left\{\begin{array}{cc} 1, & if\;C_{E}\leq C_{B}-R_{s}\\ \frac{C_{B}-C_{E}}{R_{s}}, & if\;C_{B}-R_{s}<C_{E}<C_{B}\\ 0, & if\;C_{B}\leq C_{E} \end{array}\right.$$

Given the transmission rate of confidential information *R_s_*, the information leakage rate can be obtained [23]:(12)$$R_{L}=\frac{I(M;Z^{n})}{n}=(1-\Delta )R_{s}$$

Hence, the lower bound of the information leakage rate is obtained:(13)$$R_{L}=\left\{\begin{array}{cc} 0, & if\;C_{E}\leq C_{B}-R_{s}\\ h(P_{m})-h(P_{w})+R_{s}, & if\;C_{B}-R_{s}<C_{E}<C_{B}\\ R_{s}, & if\;C_{B}\leq C_{E} \end{array}\right.$$

Here, *P_m_* and *P_w_* are the BER of Bob and Eve, respectively. The lower bound of the information leakage rate of Equation (13) represents the minimum secret information which is obtained by Eve.

## 3. Numerical Result and Discussion

In this section, we use MATLAB software for numerical analysis. The system parameters are as follows: The extraction ratio is 1%, the bit rate is 10 Gbit/s, the optical wavelength is 1550 nm, the EDFA gain *G*= 20 dB, the EDFA noise index *F_n_* = 5 dB, *B_o_* = 62.43 GHz, *B_e_* = 52.5 GHz, *R_c_* = 0.8 A/W, *α* = 0.2 dB/km, *I_d_* = 2 nA, *T* = 300 K and *R* = 50 Ω. Legitimate users use optical orthogonal code (7,2,1,1). In order to meet the reliability requirement, the transmitted power must be designed to ensure Bob’s BER ≤ 10^−9^. In this case, the channel capacity of legitimate users is close to 1 bit/symbol.

Figure 4 shows the relationship between the secrecy capacity, information leakage rate and the eavesdropping distance of Eve. The transmission distance of Alice and Bob is 100 km, *R_b_ = C_B_* and *R_s_ = 0.9R_b_*. With the increase in the eavesdropping distance, the SNR of Eve will deteriorate. Hence, the secrecy capacity gradually increases. As can be seen from Figure 4, when the eavesdropping distance is 62 km, *C_S_* = *R_s_*. In a [0, 62] km link, *C_S_* ≤ *R_s_*, Eve can obtain some confidential information. In a [62, 100] km link, *R_s_* ≤ *C_S_*, Eve will not obtain any information. Therefore, the secrecy capacity can only qualitatively describe which link segment is secure and which link segment is insecure, and cannot quantitatively evaluate the security of the whole link.

In a fiber-optic communication system, it is generally impossible to guarantee that the whole link has perfect security. Therefore, it is necessary to use the information leakage rate to quantitatively evaluate physical layer security in imperfect secure links. As can be seen from Figure 4, with the increase in the eavesdropping distance, the information leakage rate remains unchanged at first, then gradually decreases, and finally reduces to zero. In the [0, 0.2] km link, the information leakage rate is equal to *R_s_*, which corresponds to secrecy capacity of 0. This distance is defined as the complete interception distance. At this time, Eve can obtain all the confidential information. In the [0.2, 62] km link, the information leakage rate is less than *R_s_*, which means that some confidential information is leaked to Eve. Hence, this distance is defined as the partial interception distance. In the [62, 100] km link, the information leakage rate is equal to 0, which means that the secrecy capacity is no less than *R_s_*. At this point, the link is perfectly secure, and no confidential information is leaked to Eve. This distance is defined as the safe transmission distance.

By calculating the information leakage rate of different eavesdropping distances, the physical layer security of the whole link can be quantitatively evaluated. For example, *R_L_* will be 0.784 bit/symbol, 0.57 bit/symbol and 0.324 bit/symbol for eavesdropping distances of 20 km, 30 km and 40 km, respectively.

Figure 5 is the information leakage rate under different optical orthogonal codes. It is shown that the information leakage rate of OOC (7,3,1,1) is lower than that of OOC (7,2,1,1). The reason for this is that, whether using OOC (7,3,1,1) or OOC (7,2,1,1), Eve can only obtain one chip pulse by using an unmatched decoder. The legitimate user can obtain three chip pulses by using OOC (7,3,1,1), while it can obtain two chip pulses by using OOC (7,2,1,1). Therefore, by using OOC (7,3,1,1), Alice can achieve reliable transmission at lower chip power. This reduces Eve’s receiving power, resulting in a decrease in the eavesdropping channel capacity.

Figure 6 shows the relationship between the information leakage rate and eavesdropping distance under different *R_s_*. With the decrease in *R_s_*, the information leakage rate decreases. This is because Eve will receive more redundant information, resulting in less confidential information. This indicates that redundant information can effectively improve the physical layer security. On the other hand, with the decrease in *R_s_*, the distance range with no information leakage increases, that is, the safe transmission distance increases.

We consider an extreme case where *R_s_* = *R_b_*, that is, the system does not use wiretap code. As shown in Figure 6, even at 100 km, Eve can obtain confidential information. The reason for this is that the transmitted information has no redundancy information. From the perspective of secrecy capacity, this shows that the whole optical fiber link is insecure. However, from the perspective of the information leakage rate, the security of the whole link can be evaluated quantitatively.

Figure 7 shows the relationship between different *R_s_* and information leakage rates at a certain eavesdropping distance *d_e_*. As can be seen from Figure 7, with the increase in the eavesdropping distance, the information leakage rate will decrease. This indicates that the information leakage rate and transmission efficiency are restricted mutually. Under a certain eavesdropping distance, when *R_s_* is less than a threshold, the information leakage rate will be equal to 0, that is, perfect secrecy will be achieved. As the eavesdropping distance increases, the threshold for perfect secrecy will increase, that is, a higher rate of confidential information can be transmitted. For example, to achieve perfect secrecy, *R_s_* should be no larger than 0.1 bit/symbol for *d_e_* = 20 km, while *R_s_* should be no larger than 0.3 bit/symbol for *d_e_* = 30 km.

## 4. Conclusions

Based on the OCDMA network using wiretap code, the information leakage rate is used as the performance metric to evaluate the physical layer security. The relationship between eavesdropping distance, confidential information rate and information leakage rate is quantitatively analyzed from the perspective of a partial security system. The results show that the information leakage rate decreases with the increase in the eavesdropping position. With the increase in the confidential information rate, the information leakage rate will decrease. It is also shown that the information leakage rate and transmission efficiency are restricted mutually.

Unlike secrecy capacity, the information leakage rate can quantitatively evaluate the security of the entire fiber link, which allows designers to have a clearer understanding of the physical layer security of communication systems. Considering practical systems, it is necessary to study the information leakage rate using finite length encoding. In future work, we will investigate the physical layer security of specific error correction code.

## Figures and Tables

**Figure 1 entropy-25-01384-f001:**
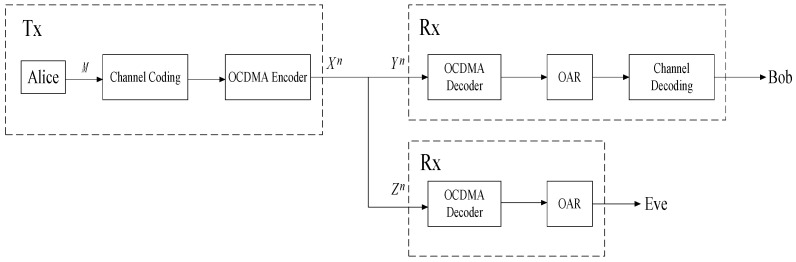
Channel model based on OCDMA using wiretap code.

**Figure 2 entropy-25-01384-f002:**
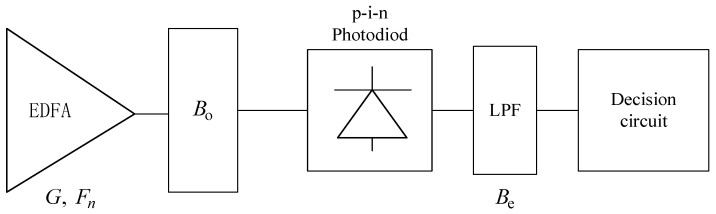
Model of optical amplifier receiver.

**Figure 3 entropy-25-01384-f003:**
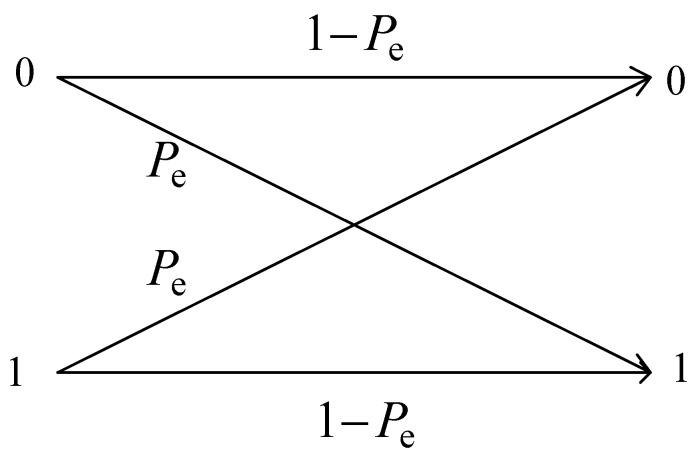
Binary symmetric channel model.

**Figure 4 entropy-25-01384-f004:**
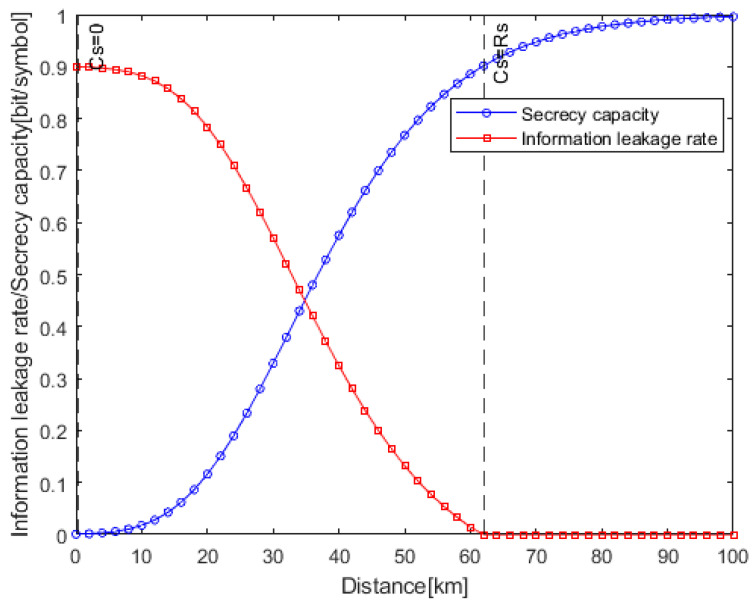
Relationship between secrecy capacity, information leakage rate and eavesdropping distance (*R_s_* = 0.9*R_b_*, *P_m_* = 10^−9^, OOC (7,2,1,1)).

**Figure 5 entropy-25-01384-f005:**
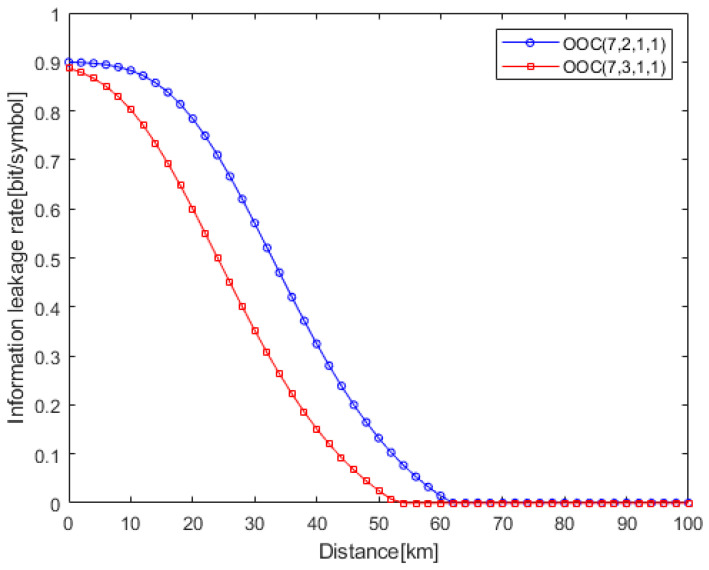
Information leakage rate under different optical orthogonal codes (*R_s_* = 0.9*R_b_*, *P_m_* = 10^−9^).

**Figure 6 entropy-25-01384-f006:**
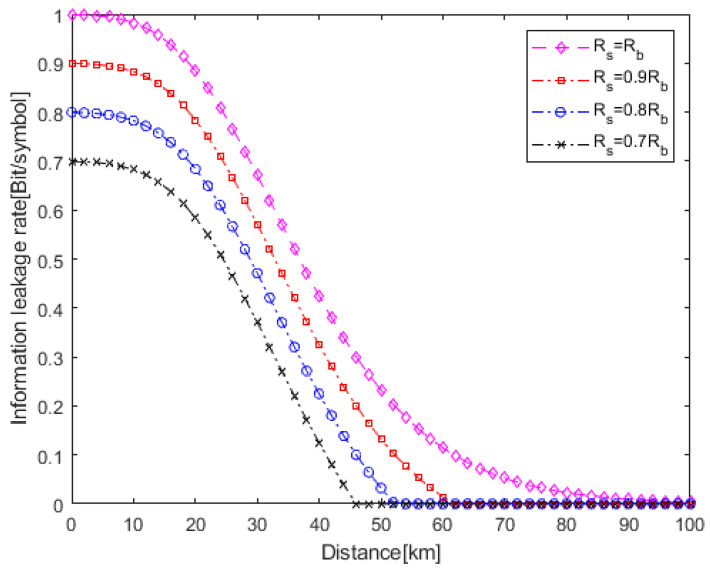
Relationship between information leakage rate and eavesdropping distance under different *R_s_* (*P_m_* = 10^−9^, OOC (7,2,1,1)).

**Figure 7 entropy-25-01384-f007:**
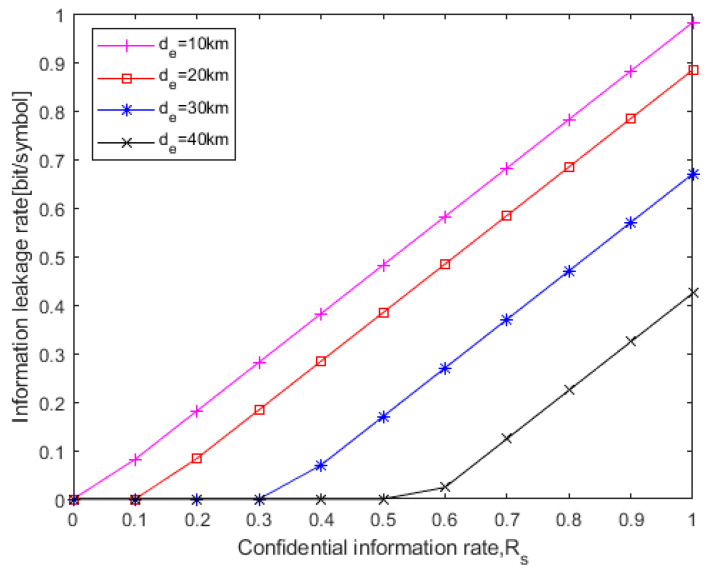
Relationship between different *R_s_* and information leakage rate at a certain eavesdropping distance (*P_m_* = 10^−9^, OOC (7,2,1,1)).

## Data Availability

Some codes generated or used during this study are available from the corresponding author upon request.

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
