# Peer review of "Information Leakage Rate of Optical Code Division Multiple Access Network Using Wiretap Code"

_entropy, 2023, doi:10.3390/e25101384_

Round 1
Reviewer 1 Report
September 8, 2023
Review of the manuscript entitled:
Information leakage rate of optical CDMA network using wiretap code
by
Rongwo Xu, Leiming Sun , Jianhua Ji, Ke Wang, Yufeng Song
Manuscript Number: ENTROPY-2585907
Introduction
The aim of the paper is to analyze information leakage rate to evaluate the physical layer security in fiber-optic wiretap channel. Applying the channel model of optical CDMA network in was shown numerically that information leakage rate is related to transmission distance, eavesdropping position, confidential information rate and optical code.
Major Comments
The results presented in the paper are interesting and of some importance but I have the following comments and concerns:
A. There is a huge difference between coding methods and encrypting methods. Encrypting methods use ciphers (etc.) and requires secret key exchange etc. just to ensure confidentiality. Why the Authors use the word security, this should be rather reliability?
B. Figure 4 presents application (graphs) of formulas derived/reported in the paper. So description (the caption) of this Figure4 should be much more informative. It should be explicitly given for which formulas the graphs are presented and the symbols that are used in these formulas should be also explicitly written with the corresponding values of parameters used.
C. The same is for Figure 5 and Figure 6.
D. The Authors applied/considered wiretap code, and theoretically analyze the information leakage rate for this code. Can in this case a kind of error correction method be applied, just do not loss information ?
Minor Comment
English could be improved slightly.
Final Comments
In my opinion this paper is interesting and of some meaning. I recommend the paper for publication, provided that the above concerns and comments will be addressed.
Minor Comment
English could be improved slightly.
Reviewer 2 Report
The authors presented a performance metric to evaluate the physical layer security based on the OCDMA network using wiretap code. They first proposed the channel model of OCDMA network using wiretap code, then theoretically analyzed the information leakage rate of fixed rate wiretap code. They discussed the relationship systematically between eavesdropping distance, confidential information rate and information leakage rate. Overall, this topic is interesting and has a certain scientific value. This manuscript is recommended to be published if the following questions can be clarified in the revision.
1. Inside the whole paper, there is no description of the range of parameters like optical fiber attenuation coefficientα, receiver responsivity Rc, optical bandwidth Bo, amplifier gain G, electron charge е, spontaneous radiation factor Nsp, Boltzmann constant, dark current.
2. Optical orthogonal code was set as (7,2,1,1). You should consider how OCDMA encoding affects the information leakage rate.
3. It is noted that the article needs careful editing by someone with expertise in technical English editing paying particular attention to English grammar, spelling, and sentence structure so that the goals and results of the study are clear to the reader.
4. All authors must also enhance the conclusion and discussing the potential future extensions of this research.
5. The experiment tools used in the design and implementation of the proposed system not clear in this work.
There are some grammar errors in the English expression of this paper. I hope the author can make further revisions to the paper in the future.
Reviewer 3 Report
Different from the qualitative measure of secrecy capacity, this paper studies the quantitative study of wiretap CDMA optical communications.
The paper is easy to follow, but since the length of the manuscript is relatively short, I recommend a shorter form, e.g., letter. More comments are:
1) For nearly every equation, especially eq. (1) - (7) on page 3, there is no reference. Potential readers may be interested in knowing how each equation is derived.
2) In eq. (3), why is it not just (1-x)P_l and instead (1-x)\frac{WP}{10^{\alphaL/10}}?
Round 2
Reviewer 1 Report
After reading the responses and actions of the Authors, I must say that the Authors satisfactorily addressed all my comments and concerns.
Therefore, I recommend the revised version of the article for publication.
Reviewer 3 Report
Thank you for addressing the previous round's questions. The reviewer does not have further questions.